# Investigating the Spatial Accessibility and Coverage of the Pediatric COVID-19 Vaccine: An Ecologic Study of Regional Health Data

**DOI:** 10.3390/vaccines12050545

**Published:** 2024-05-15

**Authors:** Amin Bemanian, Jonathan F. Mosser

**Affiliations:** 1Department of Pediatrics, University of Washington, Seattle, WA 98105, USA; jmosser@uw.edu; 2Seattle Children’s Hospital, Seattle, WA 98105, USA; 3Institute for Health Metrics and Evaluation, Seattle, WA 98105, USA

**Keywords:** COVID-19, vaccination, accessibility, pediatrics, SARS-CoV-2

## Abstract

The COVID-19 pandemic presented the unique challenge of having to deliver novel vaccines during a public health crisis. For pediatric patients, it was further complicated by the delayed timeline for authorizing the vaccine and the differences in dosing/products depending on the patient’s age. This paper investigates the relationship between the spatial accessibility and uptake of the COVID-19 vaccine in King County, WA, USA. Public data for COVID-19 vaccine sites were used to calculate spatial accessibility using an enhanced two-step floating catchment area (E2SFCA) technique. Spatial regression analyses were performed to look at the relationship between spatial accessibility and ZIP-code-level vaccination rates. The relationships of these data with other socioeconomic and demographic variables were calculated as well. Higher rates of vaccine accessibility and vaccine coverage were found in adolescent (12- to 17-year-old) individuals relative to school-age (5- to 11-year-old) individuals. Vaccine accessibility was positively associated with coverage in both age groups in the univariable analysis. This relationship was affected by neighborhood educational attainment. This paper demonstrates how measures such as E2SFCA can be used to calculate the accessibility of the COVID-19 vaccine in a region and provides insight into some of the ecological factors that affect COVID-19 vaccination rates.

## 1. Introduction

Spatial accessibility in relation to healthcare services is an important determinant of health, as it can affect a patient’s ability to receive preventative services, medications, and acute or critical care. Within the field of pediatric medicine, there have been multiple studies showing how providers and other healthcare services are not evenly distributed across populations, with disparities across socioeconomic status, racial composition, and urban/rural areas [1,2,3]. These differences in spatial accessibility have been shown to affect health outcomes as well. One study of routine immunizations among children receiving Medicaid insurance in Washington DC found that higher spatial accessibility to vaccination providers was associated with higher odds of routine vaccination completion [4]. Another study from Kenya found evidence that longer travel time to healthcare facilities for 12- to 23-month-old children was associated with lower uptake rates of the DPT3 vaccine and the full vaccination series [5]. A spatiotemporal analysis of Bacille–Calmette–Guerin (BCG) vaccination coverage of Ethiopia from 2000 to 2019 consistently demonstrated evidence of spatial clustering in the coverage [6]. Furthermore, this study showed that increased spatial distance to healthcare facilities was associated with lower rates of BCG coverage across all the study’s years. Spatial availability has been shown to affect the distribution of childhood vaccine doses as well. The mapping of human papillomavirus and tetanus/diphtheria/pertussis vaccine doses in Georgia was found to cause spatial clustering at the county level, and the authors found that public transit and the number of health department clinics were positively associated with increased vaccine doses [7]. These studies demonstrate the importance of spatial accessibility in pediatric health and vaccine utilization.

Ensuring the accessibility of care during a pandemic, however, is particularly difficult. Due to the urgent need for safe immunization options, vaccination rollout was expedited for the COVID-19/SARS-CoV-2 vaccines. In the United States, vaccinations are first authorized by the Food and Drug Association (FDA). Then, the United States Advisory Committee on Immunization Practices (ACIP) synthesizes the available data to make a recommendation to the Centers for Disease Control (CDC) about the timing and dosing of vaccinations [8]. The CDC then integrates these recommendations in order to develop the Child and Adolescent Immunization Schedule, which is used by pediatric and family medicine/general practitioner offices to deliver vaccines at children’s routine wellness appointments. Alternatively, pharmacies and, in some jurisdictions, school-based health clinics provide immunizations as well. The emergency authorization of COVID-19 vaccines for pediatric use was rolled out in a staged fashion, with the initial adult authorization (>16 years) on 20 December 2020, followed by 12- to 15-year-olds on 10 May 2021, 5- to 11-year-olds on 29 October 2021, and 6-month to 4-year-olds on 17 June 2022 [9]. The ACIP provided interim guidance during this rollout, and ultimately the CDC and ACIP announced the COVID-19 would be included in the Child and Adolescent Immunization Schedule on 10 February 2023 [10]. Due to this staged rollout and the difficulty in obtaining the initial supply of vaccines doses, some medical offices and pharmacies were not able to keep an adequate stock of COVID-19 vaccines for children and adolescents.

Additionally, the COVID-19 pandemic caused significant strain on the United States medical system and changes to patients’ health-seeking behaviors. Pediatric emergency department visits significantly decreased in 2020 relative to pre-pandemic levels and slowly rose back over the course of 2021 and 2022 [11]. Additional work has shown that emergency department visits and hospitalizations disproportionately decreased among children living in census tracts with lower socioeconomic scores, as measured by the Child Opportunity Index [12]. This decreased utilization affected preventative care services as well, with routine childhood vaccinations across all age groups decreasing during the first year of the pandemic, dropping off especially during the initial months [13].

The COVID-19 pandemic presented the unique challenge of deploying novel vaccines to curb disease spread while much of society continued to try to isolate. There have been a few studies investigating the spatial accessibility of the COVID-19 vaccine during the pandemic period. These include calculations of vaccine accessibility scores in Newark, USA using a two-step floating catchment area technique (2SFCA) [14]; Chicago, USA using a gravity-based measure [15]; and Mashhad, Iran using an enhanced two-step floating catchment area technique [16]. However, none of these papers focus on childhood or adolescent COVID-19 vaccination. For the reasons discussed earlier, the investigation into the accessibility of pediatric COVID-19 vaccine deployment is relevant to understanding how healthcare access is distributed across population areas and where the greatest needs are for vaccine providers and resources.

This paper seeks to characterize the spatial accessibility of pediatric COVID-19 vaccination sites in King County, Washington (WA), in the United States. Furthermore, it identifies associations between neighborhood-level determinants of health and vaccine accessibility to assess if there are specific communities with limited access to vaccine providers. Finally, this study investigates the relationship between vaccine accessibility and coverage, with the specific goal of testing the hypothesis that increased accessibility is associated with higher rates of vaccine uptake within communities.

## 2. Materials and Methods

The *tidycensus* R package was applied to the 2019 5-year American Community Survey (ACS) to obtain demographic and socioeconomic data [17,18]. All variables were obtained at the level of ZIP Code Tabulation Area (ZCTA). Variables obtained from the ACS included age-stratified population counts, the percentage of households living below the poverty line, the percentage of adults aged ≥25 years who graduated with a bachelor’s degree or higher, and the percentage of residents belonging to specific racial/ethnic groups. COVID-19 vaccine coverage data were obtained from Public Health-Seattle and King County (PHSKC) COVID-19 Dashboard, reflecting the latest Washington Immunization Information System data up to 5 July 2022 [19]. Coverage was defined as the percentage of children who had completed a primary COVID-19 vaccination series (two doses of either the Pfizer-BioNTech BNT162b2 or Moderna mRNA-1273 mRNA vaccines). Vaccine coverage data were stratified by age into two groups: 5- to 11-year-olds (school-age children) and 12- to 17-year-olds (adolescents). Data on younger children were not available at the time of this analysis.

Vaccine accessibility was estimated using the enhanced two-step floating catchment area (E2SFCA) technique [20]. Vaccine providers within King County were identified using the public Washington Vaccine Locator website and were stratified based on which vaccine series they provided: the adolescent/adult series (n = 434) and/or the childhood series (n = 152). Any site that was listed as a “mobile center” was excluded from the analysis due to a lack of information on whether the address corresponded to where the center distributed the vaccine or where the vaccines were stored between drives.

The steps of the E2SFCA technique are briefly described here, with an in-depth mathematical derivation included in Appendix A. First, for every ZCTA, 200 spatial points within the ZCTA’s boundaries were randomly sampled and weighted based on a population density raster from WorldPop [21]. This sampling process aimed to reduce bias from picking a single point (e.g., geometric centroid) to represent each ZCTA, which could particularly affect large, sparsely populated ZCTAs. Next, we calculated travel times and distances from each sampled point to each vaccination site using the R5R package and OpenStreetMap road network files [22]. Using these travel time distances, we then calculated the ZCTA’s accessibility score for each of the sampled points. Two sets of scores based on private automobile and public transportation times were calculated. These scores were then weighted based the percentage of households who did not own any private automobiles (from the American Community Survey) to calculate a combined score [17,18]. From this sample space, the median accessibility score for each ZCTA was selected. The derivation of the time weighting function for E2SFCA is provided in Appendix A. Two populated King County ZCTAs were excluded from the analysis. The ZCTA encompassing Vashon Island, an island municipality within King County, required a water ferry to reach Seattle and the rest of the mainland. As a result, it was a significant outlier for vaccine accessibility scores. Another ZCTA in the eastern part of King County crossed over into the neighboring Kittitas County and was excluded since we had incomplete vaccine coverage data. Additionally, several ZCTAs that had no population and only represented commercial addresses were excluded as well.

The relationship between accessibility and coverage was then assessed at the ZCTA level using spatial error regression, as determined using the *spatialreg* R package [23,24]. Ordinary linear regression was unable to be used given the spatial autocorrelation in the residuals of most of the models. Analysis was stratified by the two age groups. First, we assessed if vaccine accessibility was associated with household poverty, racial composition, and adult educational attainment at the ZCTA level. We then investigated the relationship between accessibility and coverage with several models: a series of univariable regressions assessing accessibility and the socioeconomic/demographic variables, a multivariable regression with the socioeconomic/demographic variables added as covariates with no interaction terms, and a multivariable regression that included significant interactions with vaccine accessibility. Finally, we stratified a univariable regression analysis between accessibility and coverage based on ZCTA educational attainment. We selected a threshold of 50% for adults with bachelor’s degrees (sample median: 52%) for the bifurcation. We only conducted univariable regressions for the stratified analysis, as the number of ZCTAs in each stratified model was low and multivariable analyses would overfit.

## 3. Results

### 3.1. Descriptive Analyses of Vaccine Accessibility and Coverage

Figure 1 shows the location of vaccination sites across King County for both school-age and adolescent groups. There were nearly three times as many adolescent than school-age vaccination sites (434 vs. 152). For both groups, most sites were clustered on the western side of the county around Seattle and the immediate neighboring suburbs. The eastern half of King County, WA, is more mountainous and less densely populated. Notably, there were almost no school-age vaccination sites listed in the Vaccine Locator for eastern King County and only a small number of adolescent sites.

Descriptive summaries of the covariates of interest and vaccine accessibility and coverage are included in Table 1. There was a significant difference in vaccine accessibility between adolescents and school-aged children across ZCTAs. The mean ZCTA-level difference in school-age vs. adolescent vaccine accessibility was 87.3 fewer vaccine sites per 100,000 children for school-age children (paired *t*-test *p* < 0.001). Similarly, there was a significant mean ZCTA difference in vaccine coverage, with school-age children having 21.4% lower coverage than adolescents (paired *t*-test *p* < 0.001). Adolescent accessibility was correlated with school-age accessibility, with an R of 0.90 across ZCTAs. Similarly, adolescent and school-age coverage were correlated, with an R of 0.786.

Vaccine accessibility figures are shown in Figure 2. Across ZCTAs, the interquartile range was from 112.5 to 141.7 sites per 100,000 individuals for adolescents and 31.6 to 42.9 sites per 100,000 individuals for school-age children. The highest areas of accessibility were in southern Seattle and the nearby southeastern suburbs of Kent, Renton, and Mercer Island. Notably, the general patterns of high- and low-accessibility ZCTAs are shown to be preserved across both age groups. Figure 3 shows vaccine coverage across the region. The interquartile range for adolescents was from 64.3% to 95% (the maximum reported value by the PHSKC website), and for school-age children it was from 35.2% to 69.3%. Again, coverage was higher in Seattle and its neighbors compared to the eastern half of the county. However, compared to accessibility, the ZCTAs with the highest coverage were found in northern Seattle and the near eastern suburbs of Bellevue, Kirkland, and Redmond. Two of the geographically largest ZCTAs in the county did not have complete reports of vaccine coverage due to censoring from PHSKC in response to low population counts. These locations represent relatively rural mountain communities that are fairly isolated from the majority of the population in the western portion of King County.

### 3.2. Regression Analysis between Vaccine Accessibility and Coverage

Regression analyses exploring factors associated with vaccine accessibility at the ZCTA level are shown in Table 2. The relationships identified are the same across school-age and adolescent vaccine accessibility. The percent of adults with a bachelor’s or higher degree within a ZCTA and the percent of Asian residents were positively associated with the ZCTA’s accessibility. There were no other significant predictors of accessibility in either age group. For both the school-aged and adolescent accessibility models, the spatial autoregression term was significant, suggesting that there was significant spatial clustering that was not explained by the factors included in our models.

The regression analyses used to evaluate factors potentially associated with vaccine coverage at the ZCTA level are shown in Table 3 and Table 4 for school-age children and adolescents, respectively. Among school-age children, vaccine accessibility was found to be a significant predictor of increased vaccine coverage in the univariable analysis, but it was not significant in the multivariable models. Other significant predictors in the univariable analysis included positive associations with vaccine coverage for the percentage of Asian residents and the percentage of adults with a bachelor’s degree or higher for education. Negative associations with vaccine coverage were found for the percentage of American Indian/Alaska Native residents, the percentage of Hispanic residents, and the percentage of households below the federal poverty line. In the multivariable analysis, the relationships for the percentage of Asian residents, the percentage of American Indian/Alaska Native residents, the percentage of adults with bachelor’s degrees or higher and the percentage of households below the poverty line remained significant and consistent in terms of direction, although for all three the magnitude of the effect decreased. The percentage of Hispanic residents was found to be significantly positively associated with accessibility in the multivariable model, which was inconsistent with its univariable effect. Finally, an interaction term between vaccine accessibility and the percentage of adults with a bachelor’s degree was added, because the percentage of adults with a bachelor’s degree was a significant predictor of accessibility. The interaction was found to be non-significant and the model fit was not improved between the two models (no interaction model AIC of 546 vs. interaction model AIC of 546.8). In all of the school-age models, the spatial autocorrelation coefficient was significant.

For the adolescent group, vaccine access was also positively associated with vaccine coverage in the univariable analysis. It was not significantly associated with coverage in the multivariable adjusted model without interactions, but it became significant again once interaction terms for vaccine access and the percentage of bachelor’s degrees were added to the model. The percentage of bachelor’s degrees was found to be significantly associated with vaccine coverage in the univariable analysis and in both multivariable models. There was a significant negative interaction term between vaccine access and the percentage of residents with bachelor’s degree. Therefore, this model predicts vaccine coverage increases with both vaccine access and percentage of bachelor’s degree, but the predicted increase will be less than the additive increase in the access and bachelor’s degrees terms alone. The AIC of the multivariable model decreases with the addition of this term (579.0 to 572.4), which suggests that this addition improves the model’s fit.

The percentage of households below the poverty line and percentage of American Indian/Alaska Native residents were negatively associated with adolescent vaccine coverage in the univariable analysis, but these relationships were non-significant in both multivariable models. The percentage of Asian residents was significantly and positively associated with adolescent vaccine coverage in the univariable analysis and both multivariable models. The percentage of Hispanic residents was not significantly associated with coverage in the univariable analysis, but was found to be significantly associated with increased coverage in both multivariable models.

Due to the significant interaction between vaccine access and percentage of bachelor’s degrees among adults, the univariable analysis between vaccine access and coverage was stratified based on the percentage of bachelor’s degree. These results are shown in Table 5. Vaccine accessibility was only significantly associated with coverage among adolescents living in ZCTAs with <50% bachelor’s degrees among adults. There was no significant relationship between accessibility and coverage in the ≥50% bachelor’s degrees ZCTAs for either age group.

## 4. Discussion

In this study, we used the E2SFCA method to estimate pediatric COVID-19 vaccine accessibility in King County, WA, and investigated differences in terms of access and coverage across different age groups and in relation to other socioeconomic and demographic variables. At the time of writing this article, there were several papers recently published mapping COVID-19 vaccine accessibility, but none that focused specifically on pediatric populations [14,15,16]. We showed that the areas of highest accessibility within King County were located in Seattle and the immediately adjacent suburbs. In particular, the southernmost neighborhoods of Seattle and the suburbs of Renton, Newcastle, and Mercer Island had the highest accessibility scores. This was likely due to the confluence of multiple interstate highways (I-5, I-90, and I-405) in this area, which decreased the estimated travel times to vaccination sites.

While the relative patterns of vaccine accessibility are similar between the two age groups, the magnitudes were found to be significantly different, with most tracts having school-age accessibility scores that were between 1/3rd and 1/4th of the adolescent score. This reflected the lower number of vaccination sites providing school-age vaccines. Similarly, vaccine coverage was shown to have the same relative patterns between the two age groups, but there was a mean difference of −21.5% between the coverage rates of adolescents and school-age children. Unfortunately, this was in line with US national trends on pediatric vaccination. By the end of August 2022, 60.4% of US adolescents had completed the primary COVID-19 vaccination series, while only 30.5% of school-age children had [25]. This likely in part reflected the differences in eligibility and authorization timeline between the two age groups, with the vaccine for 12- to 15-year-olds authorized on 5/10/2021 in the United States, while the vaccine for 5- to 11-year-olds was authorized on 10/29/2021. Furthermore, the dosage differences between the school-age group and the adolescent/adult vaccine may have required vaccination sites to develop different vaccine storage protocols and have additional staffing requirements in order to meet the additional demand.

Given that pediatric COVID-19 vaccination lagged compared to adult coverage, it was important to identify significant predictors of vaccination coverage. The most notable difference between the school-age and adolescent groups was that vaccine access was not associated with coverage in school-age children in any of the multivariable analyses, while it was associated with coverage for adolescents after adjusting for an interaction term with our measure of a neighborhood’s educational attainment (percentage of adults with bachelor’s or higher degrees). Interestingly, the relationship between accessibility and coverage appeared to be limited to ZCTAs with lower levels of educational attainment, as shown in the stratified analysis presented in Table 4. In both age groups, average educational attainment was the single strongest positive predictor of both coverage rates and accessibility scores at the ZCTA level. There is evidence showing that higher levels of educational attainment are associated with higher interest in COVID-19 vaccination in several survey studies in the United States and several other countries [26,27,28,29]. One hypothesis for explaining why accessibility is a less important predictor in neighborhoods with higher levels of educational attainment may be that caregivers who have higher levels of educational attainment may have a greater surplus of time and more flexible work schedules to take their children to be vaccinated. Therefore, the likelihood of these children being vaccinated could be less dependent on the spatial accessibility of vaccination sites. Alternatively, accessibility was correlated with educational attainment for both age groups. It may be that there is a specific threshold of spatial accessibility that is necessary for high rates of vaccination coverage, and the likelihood that a ZCTA has sufficient accessibility is better predicted by educational attainment. Accessibility beyond that level has diminishing returns, and therefore resulted in a statistically non-significant relationship in our models. The percentage of household poverty was negatively associated with school-age vaccination rates, which was consistent with previous work looking at vaccination coverage and healthcare utilization during the pandemic [12,30].

Racial and ethnic disparities in COVID-19 outcomes and vaccination were major health equity concern throughout the COVID-19 pandemic. It is important to note that we use the terms “race” and “ethnicity” as they are defined by the US Census Bureau. Our usage of these categories is not to suggest that there are biologic differences between these groups. Instead, the goal of including racial composition variables was to look for the possible effects of structural racism and segregation, as they have often been shown to be major social determinants of health. One previous study looking at COVID-19 vaccination coverage in Texas found that ZCTAs with percentages higher than the median of Black, Hispanic, and American Indian/Alaska Native residents had significantly lower rates of vaccine coverage [30]. Our analysis found that the percentage of AI/AN residents was associated with lower rates of school-age vaccine coverage in the multivariable model without interactions, but that this relationship was non-significant in the model with interactions. Meanwhile, the percentages of Asian, Hispanic, and NH/PI residents were all associated with higher levels of vaccine coverage in the multivariable analysis.

The positive relationship between percent Hispanic residents and vaccine coverage in the multivariable models is difficult to interpret given the inconsistent estimated effects between the univariable models, where it was non-significant in the adolescent group and negatively associated in the school-age group, and the multivariable models, where it was positively associated with the groups. This may be an issue related to multicollinearity, although the variance inflation factor (VIF) for the percentage of Hispanic residents in both multivariable models was reassuring (1.656 for school-age children and 1.594 for adolescents) Therefore, we re-ran regression models, using the percent Hispanic residents as a predictor and adding in one additional variable each time, to assess which additional covariates were responsible for the observed effect. For school-age children, the addition of variables relating to the percent of vaccination for NH/PI and Asian demographics both made the effect of the percentage of the population that was Hispanic non-significant, and the addition of percent adults with a bachelor’s degrees reversed the trend of the percent Hispanic coefficient. Similarly, for adolescents, adding percent adults with a bachelor’s degree caused the coefficient on the percent Hispanic covariate to become positive and significant. This would suggest that, when controlling for ZCTA-level educational attainment, there may be a positive association between the Hispanic percentage of the population and vaccine coverage. In the event of instability in the effect direction, however, the results of the multivariable regression should be interpreted with caution. Moreover, the univariable analysis suggests that ZCTAs with higher percentages of Hispanic residents have lower vaccine coverage. This result may better reflect the real-world implications amongst residents of those ZCTAs: regardless of whether coverage is higher than might be expected for a given level of educational attainment, persons living in locations with lower vaccine coverage will still be at higher risk of a disease.

Furthermore, it is important to highlight that this is an ecological study, and therefore the analysis occurs at the level of ZCTAs rather than individuals. Therefore, the relationships found for racial composition should not be interpreted as equivalent relationships for individuals of that particular race, as this interpretation would represent an ecological fallacy. To illustrate this point, the ecological relationships between the racial/ethnic composition of a ZCTA and vaccine coverage, shown here, differ from published individual-level data on race and vaccination by PHSKC on their COVID-19 vaccination dashboard [19]. Vaccine coverage among AI/AN individuals across all ages (including adults) in King County was >95% in July 2022, while Hispanic and Black individuals had lower coverage rates of 70.1% and 76.7%, respectively. This difference with our ZCTA-level results highlights the difference between ecological and individual relationships. It is also important to highlight the potential effect of measurements with small numbers with limited variability. Within ZCTAs, the AI/AN and NH/PI percentages were low across the region with a mean percentage of less than 1%. As such, these variables’ relationships are at higher risk of being biased by influential observations.

This study does have several important limitations. This is a cross-sectional analysis, which limits our ability to establish any causal inferences. As with any spatial analyses, the modifiable areal unit problem may affect our ability to properly identify relationships [31]. Spatial relationships can be biased by how administrative units such as ZCTAs are drawn because the boundaries affect the aggregation of data. ZCTAs were the only sub-county level of data we were able to access in terms of vaccine coverage. We also limited our study to vaccination sites within King County due to our access to vaccine coverage data, and we did not account for intercounty travel when calculating accessibility. Another limitation was that our measures of vaccine accessibility were estimates based on assumptions of travel and healthcare utilization behaviors. We formulated them using existing street network data, publicly available vaccine site data, and the existing literature and data on healthcare travel behaviors. The severe disruptions of the COVID-19 pandemic to day-to-day travel and the staggered nature of vaccine rollouts made it difficult to predict the lived experience of the individuals living in these neighborhoods in terms of accessing the vaccine. Finally, we were not able to account for the effects of the mobile and school vaccination sites, tribal sites, and community vaccine drives that were utilized by King County to help provide the vaccine [32]. These vaccine drives outside of traditional clinics and pharmacies were an important tool used by PHSKC, other health agencies, and community leaders to help expand coverage to specific communities, especially to communities that have been historically underserved [33,34]. Additionally, it would have been interesting to know if local the incidence, prevalence, or mortality of COVID-19 affected vaccine uptake. Seattle and Washington State experienced one of the first documented outbreaks of COVID-19 in the United States, and the peak rates of case incidence and hospitalizations were seen in January 2022 [35]. Unfortunately, we do not have access to ZCTA-level incidence or mortality data, and so it is not possible to test if local variations in disease burden affected vaccine coverage. There are also other factors that may have affected vaccine demand and delivery that we could not measure such as pre-pandemic routine immunization coverage rates, attitudes towards vaccination/healthcare, exposure to pro- and anti-vaccine messaging across media sources, and the level of vaccine supply at individual vaccination sites. However, our primary point of interest was vaccine accessibility, and we believe that it is important to describe the relationships we found in our analysis, even if other factors could not be accounted for.

Despite these limitations, we believe this analysis can be useful when evaluating pediatric COVID-19 vaccination access in order to help guide public health activities to improve rates of the pediatric population who are up to date with recommended COVID-19 vaccinations. The levels of vaccine access and coverage for the school-age population were significantly lower than those for adolescents. More vaccination sites are needed to provide doses across the full age range of pediatric patients. Furthermore, there should be targeted efforts in neighborhoods with lower levels of educational attainment and higher rates of household poverty in order to increase vaccine coverage in these ZIP codes. These also may be areas that would benefit the most from future mobile vaccination sites as they also had the lowest accessibility scores. This study evaluated several publicly available community-level predictors of vaccine access and coverage. However, this cannot substitute for a nuanced community-driven understanding of vaccine perception and barriers to access. It is important for researchers, healthcare providers, public health organizations, and health systems to partner with local communities to ensure high and equitable coverages of vaccines. Ultimately, increasing COVID-19 vaccination rates and re-vaccinating with updated formulations among children remains an important priority for pediatric healthcare providers and public health workers seeking to help prevent future waves and outbreaks.

## 5. Conclusions

This paper demonstrates how E2SFCA can be used to assess the accessibility of a vaccine over a large geographic area. Additionally, it is the first paper to calculate the accessibility of the pediatric COVID-19 vaccine in particular, capturing the pattern of access during the rollout of the vaccine to pediatric offices and pharmacies. We show that vaccine accessibility is associated with coverage, but that this relationship seems to be affected by other demographic variables, such as educational attainment and poverty. We believe that the findings of these paper can be helpful for public health agencies and healthcare organizations in relation to the distribution of future vaccines.

## Figures and Tables

**Figure 1 vaccines-12-00545-f001:**
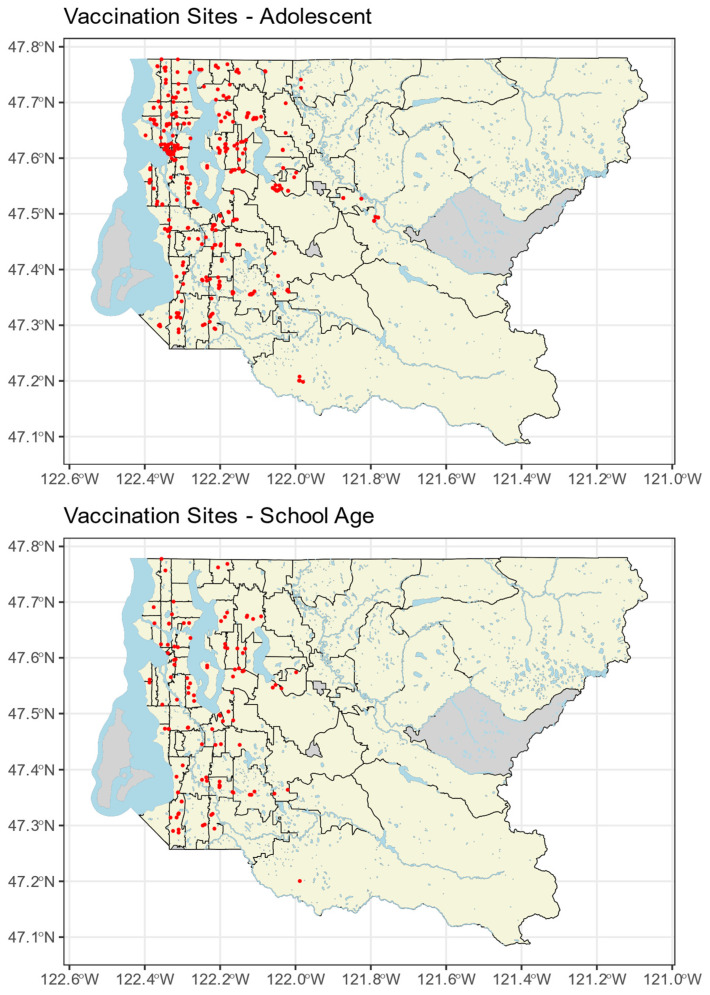
Vaccination sites within King County separated by age. Each red dot represents location of one vaccination site. Black lines indicate ZCTA boundaries and blue lines/shapes indicate bodies of water. Gray ZCTAs were not included in the analysis.

**Figure 2 vaccines-12-00545-f002:**
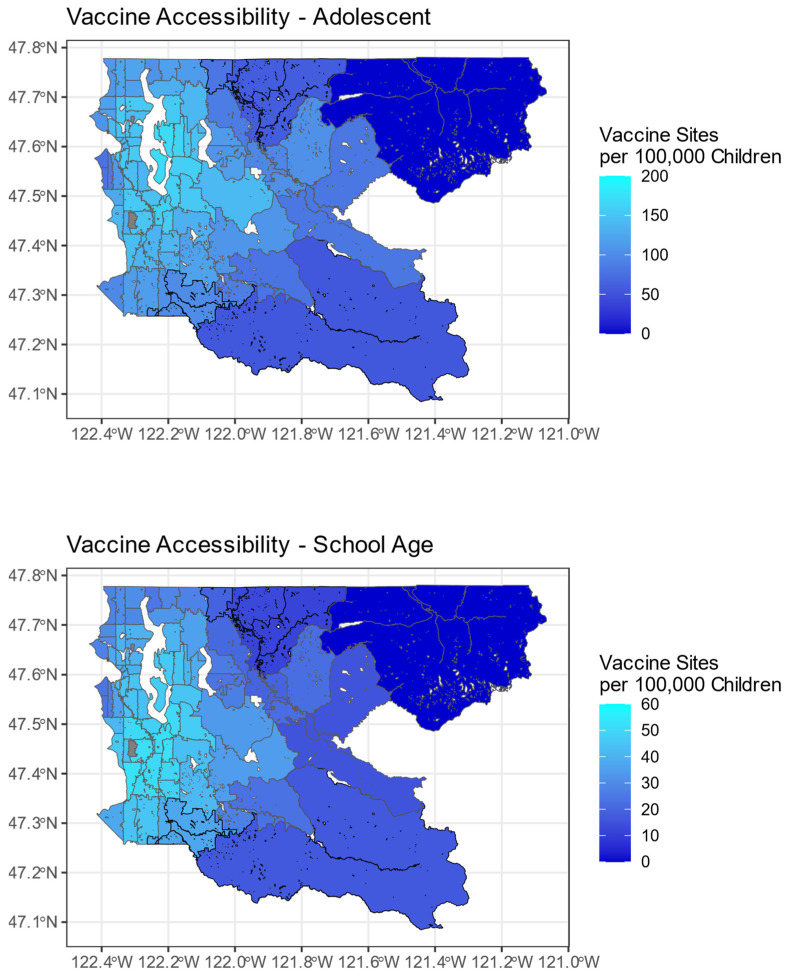
Vaccine accessibility by ZCTA in King County. Note the significantly different scale magnitudes (maximum score of 200 for adolescents vs. 60 for school-aged children) employed to better compare the trends across maps. Gray ZCTAs were excluded due to zero population (University of Washington Seattle Campus Buildings and Seattle–Tacoma International Airport).

**Figure 3 vaccines-12-00545-f003:**
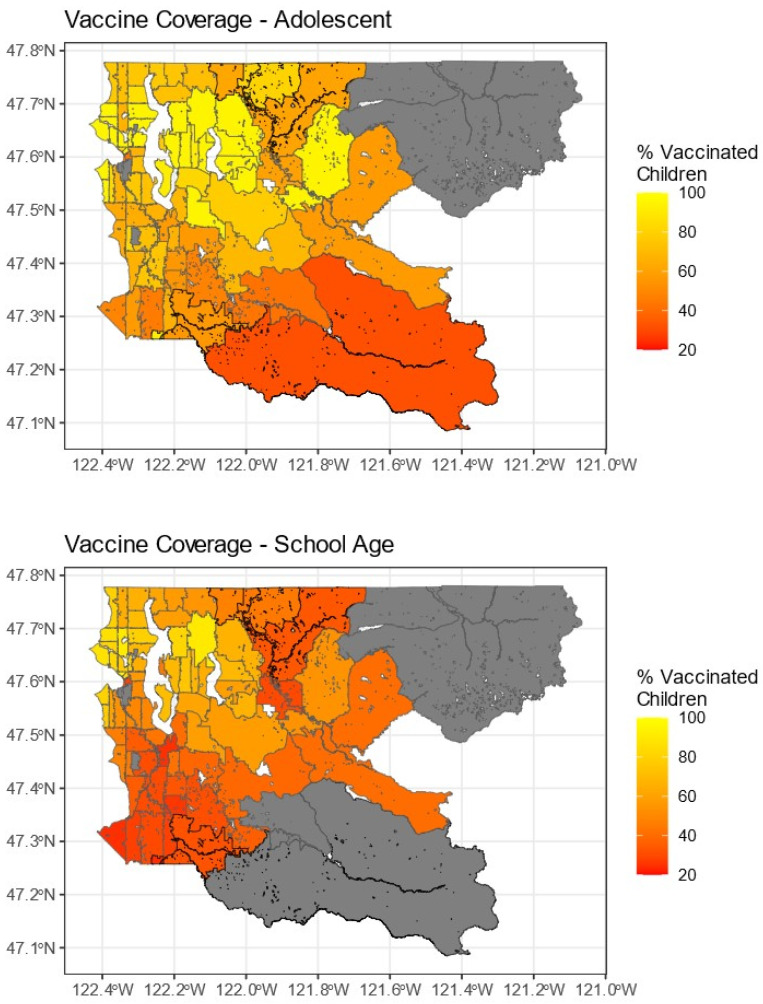
Vaccine coverage by ZCTA in King County. Gray ZCTAs were either excluded due to no population or censored due to less than 10 residents of the age group.

**Table 1 vaccines-12-00545-t001:** Descriptive statistics of King County ZCTA demographics, vaccine accessibility, and vaccine coverage.

	ZCTA Mean	ZCTA SD
**Population Covariates**		
% American Indian/Alaska Native residents	0.4	0.6
% Asian residents	16.6	10.5
% Black residents	5.7	6.1
% Hispanic residents	10.3	8.3
% Native Hawaiian/Pacific Islander residents	0.7	1.2
% Households below the poverty Line	4.9	3.6
% Adults with bachelor’s degree or higher	51.6	20.4
% Households with no personal automobiles	10.1	10.8
**Vaccine Accessibility (accessible sites per 100,000 children)**		
Adolescent (12–17 years)	122.9	34.2
School-age (5–11 years)	35.7	11.7
**Vaccine Coverage (percent of children with primary series completed)**		
Adolescent	75.8	17.0
School-age	54.4	20.6

**Table 2 vaccines-12-00545-t002:** Regression models of vaccine accessibility. Lambda is spatial autoregressive coefficient in the spatial error model. Terms with significance at *p* < 0.05 are in bold.

	School-Age Accessibility			Adolescent Accessibility		
	Coeff	95% CI	*p*-Value	Coeff	95% CI	*p*-Value
% American Indian/ Alaska Native residents	0.642	(−1.718, 3.001)	0.594	4.481	(−3.798, 12.76)	0.289
% Asian residents	**0.218**	**(0.069, 0.367)**	**0.004**	**0.735**	**(0.682, 0.788)**	**0.007**
% Black residents	0.071	(−0.301, 0.443)	0.707	0.649	(−0.629, 1.927)	0.32
% Hispanic residents	0.133	(−0.053, 0.319)	0.161	0.584	(−0.080, 1.248)	0.085
% Native Hawaiian/ Pacific Islander residents	0.524	(−0.919, 1.967)	0.477	1.236	(−3.840, 6.312)	0.633
% Households below poverty line	−0.186	(−0.504, 0.131)	0.252	−0.538	(−1.692, 0.616)	0.361
% Adults with bachelor’s or higher degree	**0.158**	**(0.038, 0.278)**	**0.010**	**0.825**	**(0.407, 1.242)**	**<0.001**
Lambda	**0.897**	-	**<0.001**	**0.818**	-	**<0.001**
AIC	504.6	-	-	699	-	-

**Table 3 vaccines-12-00545-t003:** Regression models of vaccine coverage for school-age children. Lambda is spatial autoregressive coefficient in spatial error terms. Terms with significance at *p* < 0.05 are in bold. ^†^ Univariable Lambda and AIC are for the vaccine access models.

School Age Vaccination Coverage	Univariable			Multivariable			Multivariable with Interaction Term		
	Coeff	95% CI	*p*-Value	Coeff	95% CI	*p*-Value	Coeff	95% CI	*p*-Value
Vaccine access	**0.686**	**(0.196, 1.176)**	**0.006**	0.074	(−0.267, 0.415)	0.67	0.505	(−0.326, 1.336)	0.234
% American Indian/ Alaska Native residents	**−6.541**	**(−12.401,** **−0.681)**	**0.029**	**−4.106**	**(−8.008,** **−0.204)**	**0.039**	−3.546	(−7.509, 0.417)	0.079
% Asian residents	**0.485**	**(0.14, 0.83)**	**0.006**	**0.412**	**(0.155,** **0.669)**	**0.002**	**0.37**	**(0.105,** **0.635)**	**0.006**
% Black residents	−0.46	(−1.291, 0.371)	0.278	0.222	(−0.354, 0.798)	0.457	0.215	(−0.365, 0.795)	0.467
% Hispanic residents	**−0.719**	**(−1.403,** **−0.035)**	**0.039**	**0.652**	**(0.138,** **1.166)**	**0.013**	**0.611**	**(0.090,** **1.132)**	**0.022**
% Native Hawaiian/ Pacific Islander residents	−2.571	(−6.005, 0.863)	0.142	−0.733	(−3.059, 1.594)	0.537	−0.873	(−3.207, 1.461)	0.463
% Households below poverty line	**−1.601**	**(−2.469,** **−0.732)**	**<0.001**	**−1.145**	**(−1.855,** **−0.435)**	**0.002**	**−1.158**	**(−1.989,** **−0.327)**	**0.001**
% Adults with bachelor’s degree or higher	**0.909**	**(0.720, 1.097)**	**<0.001**	**0.811**	**(0.580,** **1.042)**	**< 0.001**	**1.127**	**(0.547,** **1.707)**	**<0.001**
Access–bachelor’s degree interaction term	-	-	-	-	-	-	−0.009	(−0.025, 0.007)	0.26
Lambda	**0.845** ^†^		**<0.001**	**0.781**		**<0.001**	**0.752**		**<0.001**
AIC	600.0 ^†^		-	546.0		-	546.8		-

**Table 4 vaccines-12-00545-t004:** Regression models of vaccine coverage for adolescents. Lambda is spatial autoregressive coefficient in spatial error term. Terms with significance at *p* < 0.05 are in bold. ^†^ Univariable Lambda and AIC are for the vaccine access models.

Adolescent Vaccination Coverage	Univariable Models			Multivariable Model			Multivariable with Interaction Term		
	Coeff	95% CI	*p*-Value	Coeff	95% CI	*p*-Value	Coeff	95% CI	*p*-Value
Vaccine access	**0.231**	**(0.070,** **0.391)**	**0.005**	−0.061	(−0.166, 0.045)	0.261	**0.385**	**(0.081,** **0.689)**	**0.013**
% American Indian/ Alaska Native residents	**−7.365**	**(−14.241,** **−0.489)**	**0.036**	−2.523	(−6.815, 1.791)	0.251	−0.715	(−4.849, 3.419)	0.735
% Asian residents	**0.592**	**(0.202,** **0.982)**	**0.003**	**0.351**	**(0.080,** **0.623)**	**0.011**	**0.283**	**(0.034,** **0.532)**	**0.026**
% Black residents	−0.302	(−1.111, 0.507)	0.465	0.394	(−0.173, 0.964)	0.176	0.058	(−0.510, 0.626)	0.842
% Hispanic residents	−0.754	(−1.532, 0.024)	0.057	**1.042**	**(0.373,** **1.714)**	**0.003**	**0.823**	**(0.157,** **1.489)**	**0.016**
% Native Hawaiian/ Pacific Islander residents	2.073	(−1.820, 5.966)	0.296	**4.175**	**(1.739,** **6.623)**	**<0.001**	**4.777**	**(2.476,** **7.078)**	**<0.001**
% Households below poverty line	**−1.228**	**(−2.353,** **−0.103)**	**0.032**	−0.816	(−1.836, 0.209)	0.118	−0.656	(−1.628, 0.316)	0.186
% Adults with bachelor’s degree or higher	**0.629**	**(0.476,** **0.781)**	**<0.001**	**0.965**	**(0.751,** **1.181)**	**<0.001**	**1.896**	**(1.271,** **2.521)**	**<0.001**
Access–bachelor’s degree interaction term	-		-	-		-	**−0.008**	**(−0.012,** **−0.004)**	**0.002**
Lambda	**0.556** ^†^		**<0.001**	−0.134		0.466	−0.217		0.24
AIC	620.9 ^†^		-	579.0		-	572.4		-

**Table 5 vaccines-12-00545-t005:** Stratified spatial regression analysis of the relationship between vaccine accessibility and coverage, with stratification based on age group and the educational attainment level of the ZCTAs. No other covariates were included in these analyses. Terms with significance at *p* < 0.05 are in bold.

	Coverage in ZCTAs with ≥50% Bachelor’s Degrees among Adults (n = 41 ZCTAs)			Coverage in ZCTAs with <50% Bachelor’s Degrees among Adults (n = 33 ZCTAs)		
	Coeff	95% CI	*p*-Value	Coeff	95% CI	*p*-Value
**Adolescent** **accessibility**	0.056	(−0.060, 0.172)	0.341	**0.239**	**(0.092, 0.386)**	**0.001**
**School-age** **accessibility**	0.502	(−0.089, 1.092)	0.096	0.188	(−0.203, 0.579)	0.356

## Data Availability

Current vaccine locations can be found using the Vaccinate WA website at: https://vaccinelocator.doh.wa.gov/?language=en (accessed on 5 July 2022) and updated vaccine coverage data can be found using the Public Health Seattle & King County website at: https://kingcounty.gov/en/legacy/depts/health/covid-19/data/vaccination.aspx (accessed on 5 July 2022) These websites do not archive previous data. The specific datasets used in the analysis can be found in our code repository athttps://github.com/abemanian/VaccineAccess (accessed on 5 July 2022).

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
