# Peer review of "Investigating the Spatial Accessibility and Coverage of the Pediatric COVID-19 Vaccine: An Ecologic Study of Regional Health Data"

_vaccines, 2024, doi:10.3390/vaccines12050545_

Round 1

Reviewer 1 Report (Previous Reviewer 1)

Comments and Suggestions for Authors

I am content with the changes that the authors have made to the previous version of this manuscript. For me, the manuscript can now be published as an article. 

Author Response

We appreciate the reviewer's comments are happy that we satisfied all of their previous concerns.

Reviewer 2 Report (Previous Reviewer 2)

Comments and Suggestions for Authors

The authors have made an effort to integrate relevant information. The results and the text are more robust, reporting the 95% confidence intervals (CI95%) which are very useful for determining the association more robustly. I believe that with these changes, they have enhanced the message and strengthened their results, making the article publishable in its current form. I understand that the high Ithenticate score is due to the previous version; if this is not the reason, necessary measures should be taken.

The only correction is that Table 3 is misaligned for "American Indian/Alaska Native residents" and it adds a column that does not exist.

Author Response

We appreciate the reviewer reviewing our paper and are happy we have addressed most of their comments. We also appreciate them identifying an issue with the table layout for table 3. We will ensure it is resolved with our next submission.

This manuscript is a resubmission of an earlier submission. The following is a list of the peer review reports and author responses from that submission.

Round 1

Reviewer 1 Report

Comments and Suggestions for Authors

This is a rather good manuscript because the research presented is novel and interesting. The research is clearly explained and I did not detect any problems with the analysis and the results.  However, the text really has to be improved, while the research is interesting this is not yet an interesting papers, therfore there is a  lot of room for improvement. I was really unsure if to ask for minor or major revisions but in the end I opted for the former. Please consider carefully my point 3) below.  

1) In the first paragraph of the introduction the authors refer to a study in Georgia, US on the spacial accessibility of vaccination health services and a study on the vaccination of children with Medicaid insurance in Washington DC. As the readership of the journal is really global I wonder if the authors might also include references to research on spacial accessibility of vaccination or pediatric health provision in other parts of the world? On the same issue, why is reference 17 not included in the introduction but only mentioned in the discussion?

2) This point follows from the previous one as I think that the authors should explain pediatric health care provision, and in particular vaccination, in the US. Please do not take for granted that readers of the journal are familiar with the workings of the US health system and the differences between states and counties within states. 

3) The section on methods and materials is well done and very clear but the authors need to put some further and sustained efforts into the Result section. At the moment it is rather difficult to read and that does not do justice to all the work that has gone into conducting the research. The maps are great but the tables need some more work. Further, the section is sort of underdeveloped, for example Figure 3 and Table 3 are just put one after the other  without explanation. Rewriting and reorganising the Results section is for me a neccessary condition for the acceptance of the manuscript for publication.

4) There is also another issue that bothers me and it is about language. Not being based in the United States I have difficulties with how the authors use the term Racial disparities (see page 10 line 257 and onwards). Perhaps this is just my own sensitivity as in the EU we tend to prefer the term Ethnic disparities

Reviewer 2 Report

Comments and Suggestions for Authors

The authors are thankful for the great work and effort to perform this study. These interesting results are relevant to the topic, and children’s vaccination is also a relevant topic. However, as the methodology is an Ecological Study, results always provide or suggest new hypotheses, but it is important to address any potential bias in the study to avoid addressing conclusions without sound results.

The methodology is clear, but the statistical approach and results show inconsistent results, particularly in univariate, multivariate, and multivariate analyses with interaction terms. It is strongly suggested that the authors consider the following.

1) Report the 95% confidence interval for any regression coefficient, which could help to understand the potential bias over a wide range. For table 2, 3 and 4.

2) The change in the association of a variable must be further explored and explained as negatively associated with vaccination coverage, and subsequently, it becomes positive. A negative coefficient was reported for the percentage of Hispanic residents in the univariate model (-0.719, p = 0.039), showing an initial negative association with vaccination coverage in school-age children. However, in the multivariate model, the coefficient was positive (0.652, p = 0.013), showing a positive association. This potentially important finding must be explained, since the change in multivariate adjustment is not in itself the reason for the phenomenon and must be explained.

3) The author reported the Akaike information criterion (AIC) values for regression models; however, it did not explain the interpretation and implication of the likelihood of the model, combining univariate, multivariate, and multivariate with interactions. Particularly, in Table 3 for the school-age vaccination coverage models, the multivariate model reported an AIC value of 546.0 and 546.8 for the model adjusted by interactions.

4) Vaccination is a common good with an individual risk, and in the face of the COVID-19 health emergency, many people, mainly older adults or those at high risk, have been vaccinated. However, as time went by, the adolescents followed and later followed the minors. In the analysis, it is necessary to include the case fatality rate, mortality, incidence, and hospitalization, which are indicators of clinical severity that could possibly explain the underlying higher prevalence of risk factors in populations with a low level of education or a greater willingness to be vaccinated. A greater density of vaccination sites only has an impact if there is a willingness to carry out the vaccination, mainly in children where there was greater reluctance to vaccinate due to possible future complications. Therefore, it is necessary to include these variables in the analysis to support our conclusions.